# Monitoring cfDNA in Plasma and in Other Liquid Biopsies of Advanced EGFR Mutated NSCLC Patients: A Pilot Study and a Review of the Literature

**DOI:** 10.3390/cancers13215403

**Published:** 2021-10-28

**Authors:** Michela Verzè, Roberta Minari, Letizia Gnetti, Paola Bordi, Alessandro Leonetti, Agnese Cosenza, Leonarda Ferri, Maria Majori, Massimo De Filippo, Sebastiano Buti, Donatello Gasparro, Rita Nizzoli, Cinzia Azzoni, Lorena Bottarelli, Anna Squadrilli, Paola Mozzoni, Marcello Tiseo

**Affiliations:** 1Medical Oncology Unit, University Hospital of Parma, 43126 Parma, Italy; mverze@ao.pr.it (M.V.); pbordi@ao.pr.it (P.B.); aleonetti@ao.pr.it (A.L.); acosenza@ao.pr.it (A.C.); lferri@ao.pr.it (L.F.); sbuti@ao.pr.it (S.B.); dgasparro@ao.pr.it (D.G.); rnizzoli@ao.pr.it (R.N.); annasquadrilli@outlook.com (A.S.); mtiseo@ao.pr.it (M.T.); 2Pathology Unit, University Hospital of Parma, 43126 Parma, Italy; lgnetti@ao.pr.it (L.G.); azzoni@ao.pr.it (C.A.); lorena.bottarelli@unipr.it (L.B.); 3Pneumology Unit, University Hospital of Parma, 43126 Parma, Italy; mmajori@ao.pr.it; 4Radiology Unit, University Hospital of Parma, 43126 Parma, Italy; mdefilippo@ao.pr.it; 5Department of Medicine and Surgery, University of Parma, 43126 Parma, Italy; paola.mozzoni@unipr.it

**Keywords:** non-small cell lung cancer, EGFR-tyrosine kinase inhibitors, liquid biopsy, resistance mechanisms

## Abstract

**Simple Summary:**

In advanced non-small cell lung cancer (NSCLC) patients, tumor tissue biopsy represents the gold standard for molecular analysis procedures. However, to achieve the necessary information, both at the time of diagnosis and progressive disease, is sometimes challenging, considering the small cancer material available. Liquid biopsy consists of a non-invasive alternative approach that owns the potential to provide useful information for molecular diagnostic. We aimed to prove the worth of liquid biopsy as plasma but also as urine and exhaled breath condensate (EBC) as the best surrogate to tumor tissue as well as to explore the molecular mechanisms that underlying the resistance to second-line osimertinib in advanced EGFR mutated NSCLC. We believe that our findings, with the PLUREX study and the review of literature, may add another brick in the wall on the use of liquid biopsy in the clinical practice in the setting of EGFR-mutated NSCLC disease.

**Abstract:**

In order to study alternatives at the tissue biopsy to study EGFR status in NSCLC patients, we evaluated three different liquid biopsy platforms (plasma, urine and exhaled breath condensate, EBC). We also reviewed the literature of the cfDNA biological sources other than plasma and compared our results with it about the sensitivity to EGFR mutation determination. Twenty-two EGFR T790M-mutated NSCLC patients in progression to first-line treatment were enrolled and candidate to osimertinib. Plasma, urine and EBC samples were collected at baseline and every two months until progression. Molecular analysis of cfDNA was performed by ddPCR and compared to tissue results. At progression NGS analysis was performed. The EGFR activating mutation detection reached a sensitivity of 58 and 11% and for the T790M mutation of 45 and 10%, in plasma and urine samples, respectively. Any DNA content was recovered from EBC samples. Considering the plasma monitoring study, the worst survival was associated with positive shedding status; both plasma and urine molecular progression anticipated the radiological worsening. Our results confirmed the role of plasma liquid biopsy in testing EGFR mutational status, but unfortunately, did not evidence any improvement from the combination with alternative sources, as urine and EBC.

## 1. Introduction

Lung cancer is the most frequent cause of cancer-related death worldwide [1]. The majority of the lung cancer diagnoses (85%) are non-small cell lung cancer (NSCLC), which comprises lung adenocarcinoma (about 60%) and lung squamous cell carcinoma (30%). In the NSCLC scenario, about 10–15% of Caucasian cases present somatic sensitizing mutations in the epidermal growth factor receptor (EGFR), for which three generations of tyrosine kinase inhibitors (TKIs) have been developed able to significantly improve patients’ survival outcomes [2]. Despite the undeniable efficacy, within 10–12 months of first- or second-generation TKI treatment, acquired resistance is achieved due to resistant mutations, that are inevitably established [3,4]. The most frequent molecular cause of resistance to first- or second-generation of EGFR TKIs is the appearance of the p.T790M and its identification is crucial to continue a target treatment with the third-generation TKI osimertinib [5,6].

Tissue biopsy represents the gold standard for molecular analysis procedures. However, the high invasiveness of this procedure, together with the objective difficulty in obtaining necessary information in case of small biopsies, made mandatory to explore alternatives methods [7]. To this end, an alternative approach consists in liquid biopsy, which refers to the use of biological fluids as a surrogate for neoplastic tissue in order to obtain useful information for molecular diagnostic [8]. The decreased invasiveness of this practice is not its only strength. Indeed, liquid biopsy takes into account tumor heterogeneity, allowing it to follow the subclonal evolution through an almost uncomplicated blood draw, compared to tissue biopsy, which provides solely a snapshot of tumor at a specific time and site [9]. Among bio-fluids that may allow liquid biopsy there are plasma or serum, urine, saliva, or exhaled breath condensate (EBC), pleural and cerebrospinal fluid. From all these sources, it is possible to obtain a wide array of tumor-derived materials, in particular cell-free tumor DNA (ctDNA). Cancer patients release a higher and variable amount of plasma ctDNA, compared to healthy individuals, shed by tumor cells as a consequence of apoptosis and necrosis processes or eventually actively secreted out of cells through exosomes. Specifically, the amount of ctDNA shedding increases with the stage and metastatic sites [7,9,10]. The great potential of ctDNA it is due to its very short half-life (approximately 1 h), that make it suitable for measuring real-time tumor burden in response to therapy. However, ctDNA in plasma is present only at low levels compared to wild-type cell-free DNA (cfDNA), often making the detection of mutations a real challenge [9]. Indeed, the tumor shedding, or rather the release of genetic material in the bloodstream, is strongly influenced by both the timing of blood draw and the patient clinical condition. Finally, the metastatic site strongly influences the accuracy of ctDNA analysis. Notably, in a pooled analysis, the diagnostic accuracy of ctDNA for the detection of both EGFR activating and T790M resistant mutations in NSCLC patients who progressed after EGFR-TKIs was significantly higher in patients with extra-thoracic compared to intra-thoracic disease [11]. To this end, the research community is focusing on alternative sources from which isolate ctDNA.

In this review, we report our experience with the PLUREX (PLasma, URine, EXhaled) study, which aimed to assess the sensitivity of EGFR mutational screening on different cfDNA sources and their potential combination, and further to explore the molecular mechanisms that underlying the resistance to osimertinib. Furthermore, we carried out a literature review of the alternatives cfDNA biological sources other than plasma.

## 2. Materials and Methods

### 2.1. Patients and Plasma/urine/EBC Samples Collection

In University Hospital of Parma, we enrolled locally advanced (stage IIIB-C) or metastatic (stage IV) EGFR mutated NSCLC patients with confirmed T790M mutation on tissue, in disease progression (PD) to first-line treatment with first- or second-generation EGFR-TKIs. The PLUREX study is a real-world experience aiming firstly to evaluate the sensitivity of mutational screening for plasma, urine and EBC specimens as well as to identify the best combination of different sources of cfDNA that maximize the sensitivity in EGFR mutations detection. Moreover, as an explorative objective, we aimed to identify novel molecular abnormalities responsible for resistance to osimertinib, through a liquid biopsy NGS approach. This study obtained the ethical approval by local Ethics Committee and all patients signed specific informed consent form before any procedure. Patients who entered the study received osimertinib (80 mg/day) as a second-line therapy until disease progression or clinical benefit. We collected plasma, urine and EBC specimens from patients before the beginning of the therapy (baseline timepoint) and every two months until PD (Figure 1).

### 2.2. Samples Processing and Analysis of EGFR Mutational Status

Eighteen ml of blood were collected in EDTA tubes and centrifuged twice for 10 min at 2000× *g* within one hour after blood drawing. About 30–50 mL of urine were collected and centrifuged twice for 10 min at 16,000× g within one hour after collection. EBC was collected with a portable TURBO-DECCS condenser after that patient breathed tidally through the mouthpiece for 30 min when the condensate temperature of the condenser was −5 °C. All samples were stored at −80 °C until analysis. At baseline and during all timepoints considered cfDNA was extracted using the QIAmp Circulating nucleic acid kit (Qiagen^®^, Valencia, CA, USA) from 2 mL of plasma and from 4 mL of urine, respectively. cfDNA from 2 mL of EBC was extracted using QIAmp DNA Investigator kit, QIAmp Mini kit and QIAmp Circulating nucleic acid kit (Qiagen^®^, Valencia, CA, USA), but unfortunately, any DNA content was recovered from this source. All liquid biopsy samples were quantified through the fluorometric Qubit^®^ assay (Thermofisher, Waltham, MA, USA). All the procedures were executed following the specific manufacturer’s instructions. At diagnosis and at progression to first-line TKIs, EGFR mutational status were assessed on tissue as part of diagnostic procedure by validated method Therascreen EGFR RGQ real-time PCR assay (Qiagen^®^, Valencia, CA, USA). The EGFR mutational analysis (del19/L858R activating mutation and T790M resistance mutation) was performed on cfDNA obtained from biological samples by a QX100 ddPCR platform, using the ddPCR Mutation Assays (BioRad^®^, Hercules, CA, USA). All the procedures for the molecular analysis have been performed following the specific manufacturer’s instructions. 

### 2.3. NGS Analysis

NGS of plasma ctDNA at PD of osimertinib treatment was performed using the AVENIO NGS-panel (Roche, Basel, Switzerland). When available NGS analysis were also performed on PD tissue samples using Solid Tumor Solution-Plus (STS-Plus) (Sophia Genetics, Saint-Suplice, Switzerland).

### 2.4. FISH Analysis

FISH (Fluorescent In Situ Hybridization) assay was performed on tissue at osimertinib resistance, when available. MET, HER2 and EGFR copy number were evaluated as a potential mechanism of acquired resistance. Samples were classified as FISH-positive following specific guidelines [12,13,14].

### 2.5. Statistical Analysis

Progression-free survival (PFS) was defined as the duration between osimertinib initiation and progression of disease or death for any cause, whichever occurred first. Time to treatment failure (TTF) was defined as the number of months after the disease is treated before the cancer spreads and the patient’s health worsens. Similarly, overall survival (OS) was calculated from the date of starting osimertinib and death for any cause or last follow-up (censored patient). The Kaplan-Meier method was employed to estimate survival outcomes (PFS, TTF, and OS) and curves were compared by using log-rank test. Chi-square tests were used to correlate plasmatic mutations levels and tumoral response. Tumor response was evaluated according RECIST criteria version 1.1. Statistical analysis was done with SPSS v25 (IBM Corporation, Armonk, NY, USA).

## 3. Results

### 3.1. Patient’s Characteristics 

From April 2017 to October 2019, 22 advanced NSCLC patients were enrolled. Two patients were excluded from the analysis, one patient due to the presence of a concomitant other tumor and one because of a rapid worsening of the clinical conditions. The median age of the analyzed cohort (20 patients) was 52.5 (range 42–67), 35% were male, 55% never smoked, and all had adenocarcinoma histology (Table 1). At diagnosis EGFR del19 and L858R were present in 12 (60%) and 7 (35%) patients, respectively. One patient (5%) had an uncommon EGFR activating mutation (G719X). Type of first-line TKI administered was gefitinib (65%), erlotinib (20%) and afatinib (15%). 

### 3.2. Analysis on Plasma and Urine cfDNA Specimens

At baseline, 11 out of 19 plasma samples were positive for the activating mutation (the patient with G719X was excluded from the analysis because the specific probe was not available at our center) and 9 out of 20 plasma samples were positive for the T790M resistance mutation. Whereas, at baseline 2 urine samples were positive both for the activating and resistance mutation. These two positive urine samples were compared with the corresponding plasma samples in order to determine if there was any correspondence between them. We observed that one plasma sample was negative for both EGFR mutations, whereas the second positive urine sample corresponded to the plasmatic one, positive for EGFR activating and T790M mutation. 

Therefore, the EGFR activating mutation detection in plasma and urine samples reached a sensitivity of 58% and 11%, respectively. While the T790M resistance mutation detection reached a sensitivity of 45% and 10% in plasma and urine samples, respectively. The combined sensitivity (plasma + urine) was 69% for the detection of the activating mutation and 55% for the T790M resistance mutation. Since EGFR mutation positivity on tissue biopsy was an inclusion criterion, the concordance between tissue and plasma/urine corresponds to sensitivity value (Figure 2). 

Patients with ctDNA positivity at least for the sensitizing EGFR mutation at baseline were considered “ctDNA shedder”. We found that shedder patients (11) had a shorter PFS (5.3 vs. 17.0 months, *p* = 0.059), TTF (7.4 vs. 21.3 months, *p* = 0.097) and OS (15.7 vs. 26.5, *p* = 0.544), compared to non-shedder population (8) (Figure 3a). Moreover, dividing shedder patients based on the “shedding type” (shedder for the only activating mutation or shedder for both activating and T790M mutation) we found a significant difference in terms of PFS, TTF, and OS. Specifically, patients who were shedder for the only activating mutation (2) had a significantly shorter PFS (1.6 vs. 5.5 months, *p* = 0.013), TTF (4.1 vs. 7.4 months, *p* = 0.012) and OS (4.1 vs. 16.7, *p* = 0.027), compared to patients who were complete shedders (9) at baseline (Figure 3b). 

### 3.3. Plasma and Urine cfDNA Monitoring 

Plasma monitoring was performed in 19 out of 20 patients for activating mutation and in the entire cohort for T790M resistance mutation. At first plasma re-evaluation (T1), two months after osimertinib therapy, we evaluated the shedding status, and we found that patients who were still positive for at least the activating mutation (5), had a significant worst PFS (3.4 vs. 14.2 months, *p* < 0.001), TTF (4.1 vs. 19.2, *p* < 0.001) and OS (4.1 vs. 26.5 months, *p* = 0.029) than patients whom ctDNA turned into negative or still remained negative (14) (Figure 4a). Besides, all patients were negative for EGFR mutations at T1 in urine samples. 

We further analyzed ctDNA plasma clearance that occurred in 11 patients positive for activating mutation at baseline. Patients that gained clearance (6) showed a better PFS (9.2 vs. 3.4 months, *p* = 0.008), TTF (15.5 vs. 4.1, *p* = 0.011) and OS (16.7 vs. 4.1 months, *p* = 0.152), compared to patients who failed to clear ctDNA (5) (Figure 4b). Notably, considering only patients that shed both activating and T790M at baseline (9/19), we observed that patients who achieved a complete clearance (5) had a better PFS (14.2 vs. 3.4 months, *p* = 0.011), TTF (16.7 vs. 3.9, *p* = 0.007) and OS (not calculable vs. 3.9, *p* = 0.188), compared to whom retained the activating mutation (4) (Figure 4c). 

We correlated ctDNA shedding status at baseline with the type and the number of metastatic sites and we did not observe significant differences (data not shown). Similarly, no statistically significant difference was found in the correlation with shedding status at the first timepoint as well as with clearance and type and number of metastatic sites (data not shown). We also correlated ctDNA shedding status at baseline with the tumor response, but we did not find any significant difference (data not shown). Besides, patients that were non-shedder at the time of T1 showed a higher probability of tumor response, although no statistically significant difference was reached. 

Patients had routinary radiological reassessment of disease and plasma as well as urine specimens were collected every two months until PD and evaluated in their dynamic changes in terms of ctDNA status (positivity or negativity for at least the activating EGFR mutation). Timing of plasma/urine progression, here defined as any increase in allelic frequency of the sensitizing EGFR mutation in relation with baseline levels, was compared with timing of PD according RECIST radiological criteria. In the analyzed cohort (19/20) we observed that plasma PD foreruns radiological PD with a median time of 2 months; remarkably, also in urine we observed positivity in three patients before the radiological PD, with a median lead time of 2 months similarly to plasma samples (Figure 5a,b).

### 3.4. Analysis on Resistance Mechanisms to Osimertinib 

At the time of the analysis, all patients experienced PD and EGFR status was assessed for all the study cohort, on plasma by ddPCR. Plasma positivity for activating mutation was observed in 14/19 (74%) cases. Plasma positivity for T790M was observed in only 2/20 (10%) patients at the moment of osimertinib failure. Two urine samples were positive for the activating mutation (11%) and 1 (5%) for the T790M resistance mutation. Urine samples positivity corresponds to plasmatic ones. 

Based on the higher sensitivity of the ddPCR technique compared to NGS one, we set out to proceed to analyze only the 14 patients, which were defined shedders at the time of PD, through ddPCR. Firstly, we observed that the detection rate with NGS analysis of the EGFR activating mutation in the cohort was 86% (12/14) and further that the T790M resistance mutation was maintained only in 2 (14%) patients (also confirmed by ddPCR). 

As putative mechanism of resistance to osimertinib as second-line therapy EGFR gene amplification in 64% (9/14) of the patient’s cohort, MET gene amplification in 29% (4/14), TP53 missense variants in 50% (7/14), SMAD4 missense variants in 21% (3/14) and EGFR p.C797S missense mutation, EGFR p.T790M missense mutation, ERBB2 gene amplification, PI3KCA and ROS1 missense variants in 14% (2/14) were found. Furthermore, although less represented, we found other putative resistance mechanisms such as KDR, NRAS, MET, FGFR2, CDK4, KIT, ALK missense variants in 7% (1/14) of the patient’s cohort (Table 2). 

Among these 14 PD patients, 6 (43%) underwent tissue re-biopsy and the analysis of putative resistance mechanism on histological or cytological biopsies revealed that 3/6 (50%) patients presented MET amplification while 1/6 (17%) patient had both HER2 and EGFR polysomy. One (17%) patient showed EGFR exon 20 insertion and the last one (17%) incurred into small cell lung cancer transformation. The concordance rate between tissue and plasma at PD was 60% (Table 2).

## 4. Discussion

We report our experience of the PLUREX study in which we tested the sensitivity of EGFR mutational screening on different cfDNA sources (plasma, urine and EBC) in order to obtain the best surrogate to the tumor tissue. In a limited NSCLC EGFR mutated T790M positive population, we confirmed the role of the plasma with a sensitivity/concordance about of 60%, showing, however, disappointing results for urine and EBC. A parallel objective of this study was aimed at reviewing the literature, reporting data on the potential of non-blood liquid biopsy platforms in the searching of EGFR molecular aberration in NSCLC scenario. 

Several studies demonstrated a high concordance in EGFR mutation status between tissue and plasma. Weber et al. [15] examined for EGFR sensitizing mutations 196 pairs of diagnostic tissue biopsy and plasma, prior to first-generation TKI treatment and showed a 91% of overall concordance. Douillard and colleagues [16] in a cohort of 1060 EGFR-positive NSCLC patients under gefitinib treatment demonstrated a concordance of 94.3% between tissue and plasma. Duan et al. [17], on 94 paired histological and plasma NSCLC patient samples, found that overall concordance of EGFR mutation status was 80%. Furthermore, from the AURA extension and AURA2 phase II studies, Jenkins and colleagues [18] screened 551 patients. They showed high agreement between tumor tissue and plasma; in particular, they obtained a sensitivity of 61%, a specificity of 79% and an overall concordance of 65% for T790M detection, while they found a 80%, 98% and 90% of sensitivity, specificity and overall concordance, respectively, for the activating mutations detection. In our experience, the sensitivity of plasma EGFR mutational status detection was 58% for the activating and 45% for the T790M mutations, therefore slightly lower than expected. This could be probably linked to the small cohort, as well as to the tight inclusion criteria with only T790M positive patients eligible. Since the very short half-life of cfDNA and the potential contamination with genomic DNA released by white blood cells represent obstacles, a possible way to improve cfDNA quality and content could be the use of alternative blood collection tubes equipped with a cfDNA preservative [19,20]. If blood draw is an almost uncomplicated procedure, urine test is a completely noninvasive as well as easy approach to obtain cfDNA. Urine specimens have proven their worth as diagnostic tests in a variety of diseases [21] and here we report studies on which urinary cfDNA showed comparable performance with plasma cfDNA (Table 3). Chen et al. [22] assessed the sensitivity of urinary ctDNA compared to plasma ctDNA, in a comparative study conducted on 150 tissue samples, along with matched plasma and urinary specimens. The overall concordance rate among tissue and urinary samples was 88%, using digital droplet PCR (ddPCR) method, while the comparison of plasma and urine ctDNA highlighted a concordance rate of 98%. Additionally, Haiying Yu [23] piloted an observational study which involved 130 NSCLC patients who received EGFR TKIs therapy. In this study, Haiying’s group registered 85.4% and 83.1% of concordance rate in the comparison of plasma and urine with histological reference, respectively. Moreover, they calculated a cumulative liquid biopsy result with 86.2% of agreement. Furthermore, the Reckamp and colleagues [24] blinded retrospective study still stresses the strength of urinary ctDNA analysis. The cohort of this study enclosed 63 NSCLC patients formerly enrolled in the TIGER-X trial. They found a sensitivity of 17% for EGFR sensitizing mutation and 72% for T790M. Interestingly in this study, when the only samples with high urine volume (90 to 100 mL) were considered, the detection of EGFR mutations were significantly higher (81% for EGFR activating mutations and 93% for T790M, respectively). Our results are in contrast, considering the low sensitivity in EGFR mutational status detection obtained, even if in a small cohort of patients. Urinary ctDNA positive rate for the sensitizing and resistance mutations were 11% and 10%, respectively. In our practice, we did not add any preservative to urine samples and we neither concentrated the entire urine volume to maximize the cfDNA content. Therefore, in the future experiments, we will try to modify the urine sample processing protocol in order to obtain a larger amount of cfDNA and hence giving us the chance to increase the detection sensitivity rate. 

Among non-blood body fluids, a source suitable to perform liquid biopsy there is saliva. Saliva collection is a non-invasive, cost-effective, and easy method. To date, few data are available about the feasibility of performing saliva test as liquid biopsy tool. Nevertheless, we bring some stimulating results found in the literature. Shanshan Ding’s group [25] analyzed EGFR activating mutations, as well as the T790M resistance mutation, in paired saliva and plasma ctDNA of 27 NSCLC patient’s cohort. Data showed 83.78% of overall concordance rate between blood and saliva ctDNA (concordance rate was calculated including 10 paired blood and saliva healthy donors’ samples). Another notably result arise from Hackner et al. [26], which investigated the feasibility to use sputum test in the detection of EGFR activating and resistance mutations on 28 NSCLC patient’s cohort using ddPCR. The concordance rate of the EGFR sensitizing mutations status between plasma and sputum samples was 71%, whereas the concordance rate of the T790M was 86%. A further study, conducted by F. Su and colleagues [27] on 37 NSCLC patients highlighted a high reliability of the saliva test; indeed, they didn’t find false positives in sputum samples and the accuracy, the specificity and the sensitivity registered were 97.1%, 96% and 90.9% respectively. However, the use of saliva as a source of cfDNA is impaired by some limitations. Mostly, the amount of analyte that could be obtained is affected by several subjective matters for instance emotional as well as mental status that are difficult to control. Therefore, cfDNA concentration is strongly heterogenous among patients and timepoints. The last source we took into account in this review of the literature is EBC. The goal of this matrix consists of samples that are representative of airway fluids and contemporary less affected by DNA contamination, proper of the oral cave. Nishii Kazuya et al. [28] tested sensitivity and specificity of EGFR mutational status on 21 EBC-ddPCR specimens and they found 26.5% and 90.3%, respectively. Smyth’s group [29] conducted an explorative study focusing on the potential of the detection of T790M in 10 paired EBC and plasma samples. In their cohort, they demonstrated that the EBC source has a greater potential for detecting the resistance mutation, than the plasma one (9 T790M EBC-positive samples vs. 7 plasma-positive samples). Unfortunately, in our experience to obtain DNA from EBC samples it was very challenging, and we could not recover any DNA content from this source although different commercial kits were employed. 

Consistently with literature, in the PLUREX study we reported that positive shedding status at diagnosis is associated with poor survival outcome [30,31]. Unexpectedly, patients that lack T790M mutation on plasma showed a worst performance, compared to patients who were complete shedder. We speculate that this finding could be an artifact, resulting from the unbalanced distribution of patients onto groups (2 vs. 9). On the other hand, the ctDNA-positive patients on urine samples were two, and noteworthy in one case the urine test proved to be more sensitive than the plasma test, where the same sample was negative. 

The disease burden as well the number or the metastatic site strongly influence the DNA release in the body fluids; probably the non-shedding status of patients at baseline was may due to the intra-thoracic or brain metastasis presence that are known to be associated to a slighter shed of DNA [11,32,33]. Even though, in our study the small size of the cohort did not allow to drawn conclusion in the correlation of ctDNA shedding status at baseline with type or number of metastatic sites. 

From the plasma and urine cfDNA monitoring, we observed that patients who retained at least the activating mutation at first evaluation had a significant worst clinical outcome, in line with the literature [34,35,36]. In addition, we demonstrated that a positive ctDNA at T1 timepoint, hence despite the osimertinib treatment, can better predict a poorer clinical outcome than baseline. We also evaluated plasmatic clearance of EGFR mutations in shedder patient’s population after two months of osimertinib administration, confirming a longer survival in the cleared patients [34,35,36]. Interestingly, when we focused on patients which were complete shedder, we observed worse clinical outcomes from groups that retained the sensitizing mutation, similarly to Ebert et al. [37]. 

Recent data suggest that some patients experienced a rise in cfDNA concentration some months before RECIST PD [38]. Our results confirm this evidence, indeed through plasma longitudinal monitoring of EGFR mutational status during all study period until PD, we found that timing of plasma progression foreruns radiological PD with a median time of 2 months. Noteworthily, we obtained the same predictive potential from urine samples, where, in three patients, we observed dynamic change over time, in particular ctDNA turned into positive before the radiological PD, with a median lead time of 2 months similarly to plasma samples. 

As an explorative objective we studied the resistance mechanisms to osimertinib. All the study cohort experienced PD and firstly, we evaluated plasma samples through the identification of the sensitizing mutation on ddPCR. As expected, almost the entire cohort (74%) was found positive on ctDNA at the time of radiological PD. Interestingly, we observed a higher percentage of T790M loss on our plasma samples in comparison with the literature [39,40]. Concerning urine samples, we report a low positivity rate in our cohort at the time of resistance to treatment. 

At the time of PD to a TKI, a tissue re-biopsy should be performed to unravel the underlying resistance mechanisms, however often this maneuver is strongly limited by the patient’s clinical conditions. In this scenario, liquid biopsy and specifically blood-based liquid biopsy, could provide the needful molecular information [41]. Indeed, in our experience at the time of osimertinib failure we could get access to only 6 re-biopsy tissue out of 14 paired-plasma samples. We only analyzed with NGS the 14 PD plasma samples with detectable activating mutation previously stated on ddPCR. We report a similar detection rate of the EGFR driver mutations. In fact, only two samples were negative at NGS and the same samples were detected at a very low frequency also in ddPCR. The most common resistance mechanisms included a copy number variation in EGFR (64%), MET (29%) and ERBB2 (14%) genes, as previously observed [42]. In our cohort, we showed the concomitant presence of T790M, C797S and EGFR amplification in two patients and our findings are consistent with what is reported in the literature. [40]. We found other variants with at lower frequency, such as SMAD4 missense variants in 21%, PI3KCA and ROS1 missense variants in 14%, KDR, NRAS, MET, FGFR2, CDK4, KIT, ALK missense variants in 7%. 

Our study limitations mainly consist of a small cohort sample size, considering that the monocentric experience with tight inclusion criteria may have limited the patient enrollment, as well as the pre-analytical sample handling i.e., the lack in using stabilizing substances could affect both the quality and the quantity of cfDNA recovery, in particular in urine and EBC samples. In addition, the lack of a comprehensive tissue biopsy availability at the time of PD inevitably limited our work in assessing the putative resistance mechanisms. This is an open issue in the literature because a standardized protocol refers to, in terms of samples centrifuge parameters, starting volume for cfDNA extraction etc., is still missing. 

## 5. Conclusions

In conclusion, our results confirmed the role of plasma liquid biopsy in testing EGFR mutational status, but unfortunately did not evidence any improvement from the combination with alternative sources, such as urine and EBC. Also, with our study, we underlined the utility of plasma ctDNA during the treatment as prognostic factor and as complementary or exclusive source to identify targetable resistance mechanisms. Finally, liquid biopsy could play a key role also in the detection of other driver-genes such as KRAS, in NSCLC patients treated with specific TKI [43].

## Figures and Tables

**Figure 1 cancers-13-05403-f001:**
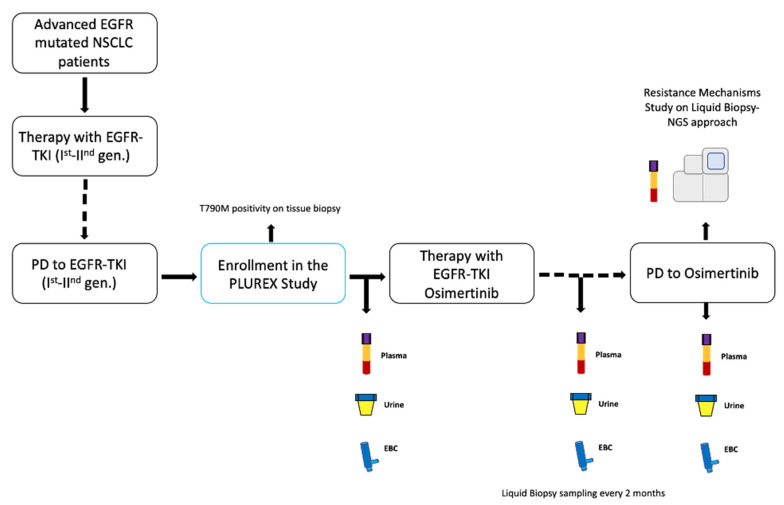
Study design. Abbreviations: TKI, Tyrosine kinase inhibitor; PD, progressive disease; gen., generation; EBC, exhaled breath condensate.

**Figure 2 cancers-13-05403-f002:**
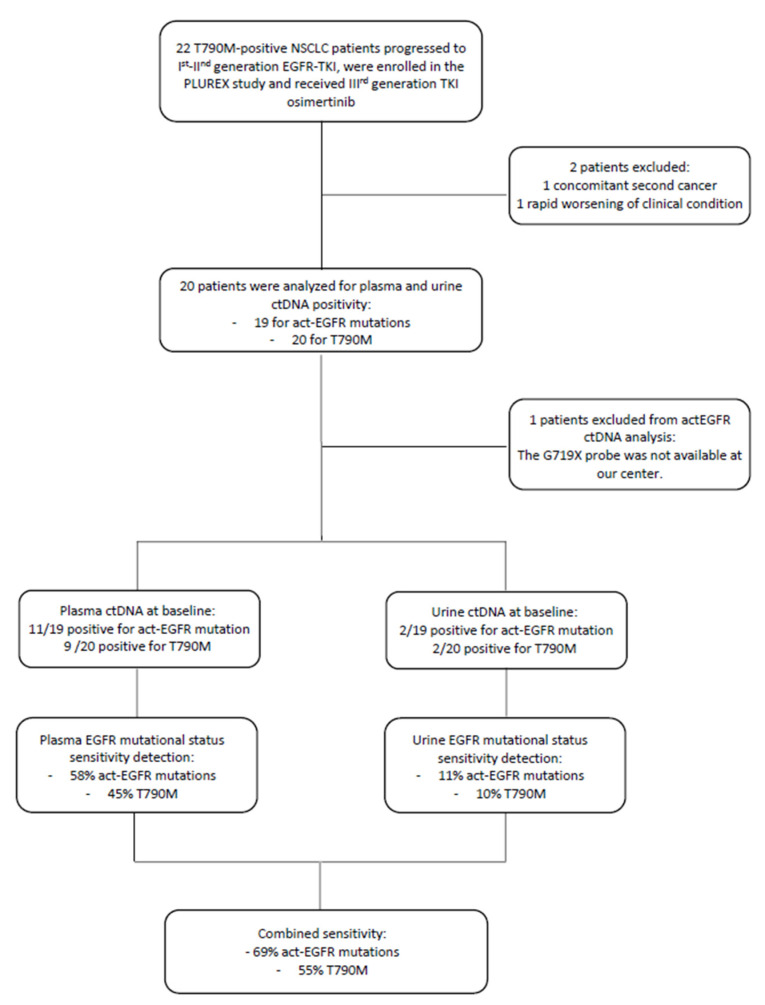
Flow-chart of patients enrolled in the PLUREX study and involved in the analysis of EGFR mutational status on plasma and urine samples. Abbreviations: ctDNA, cell-free tumor DNA; TKI, Tyrosine kinase inhibitor; gen., generation.

**Figure 3 cancers-13-05403-f003:**
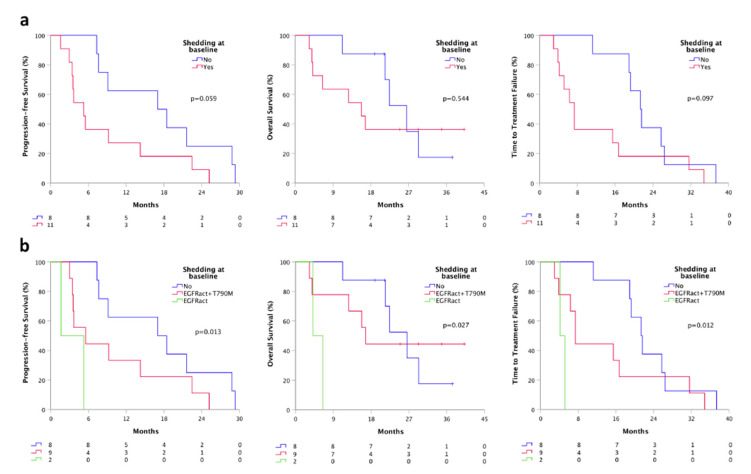
Survivals according to baseline plasmatic shedding status by ddPCR. (**a**) PFS, TTF, and OS according to the shedding status at baseline; (**b**) PFS, TTF and OS according to the shedding type at baseline. Shedder, patients with at least activating EGFR mutations detectable on plasma; non-shedder, patients with plasma sample negative for any EGFR mutation. Shedding type, shedder for the only activating mutation or shedder for both activating and T790M mutation.

**Figure 4 cancers-13-05403-f004:**
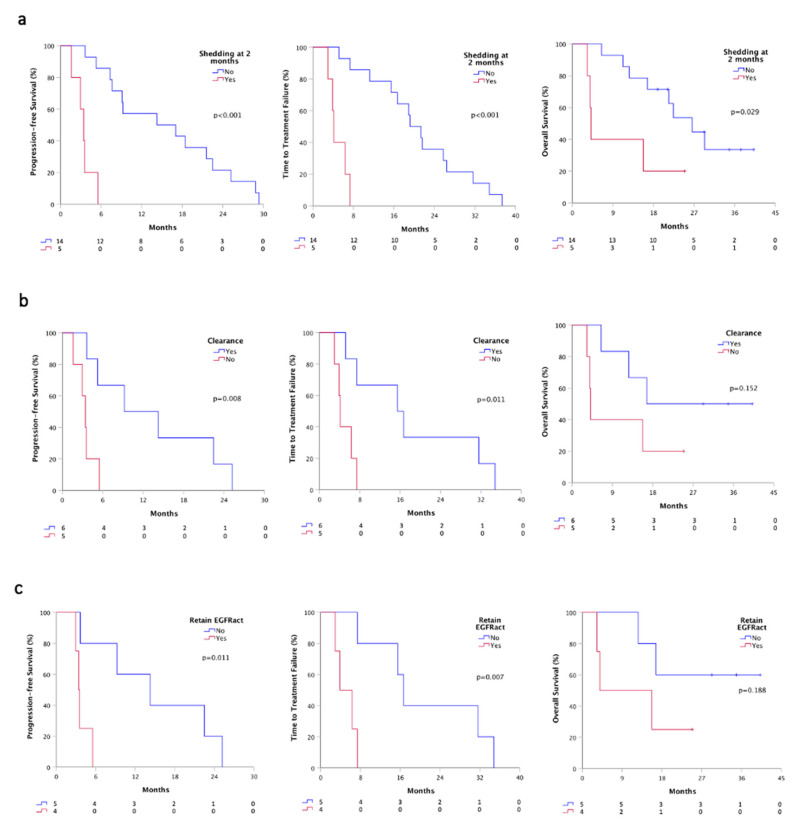
Survivals according plasmatic shedding status at T1 by ddPCR. (**a**) PFS, TTF and OS according to the shedding status at T1 time-point; (**b**) PFS, TTF and OS according to plasma clearance at T1 time-point; (**c**) PFS, TTF and OS according to the retained EGFR activating mutation. Shedder, patients with at least detectable activating EGFR mutations on plasma; non-shedder, patients with plasma sample negative for any EGFR mutation. Clearance, ctDNA positive patients at baseline that turned into negative at T1; Retained EGFRact, EGFRact and T790M positive patients at baseline that maintain activating mutation at T1.

**Figure 5 cancers-13-05403-f005:**
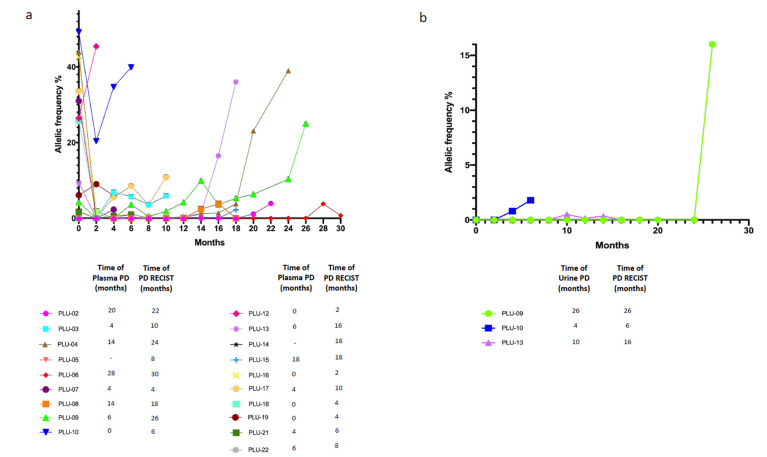
Dynamic monitoring of cfDNA status. (**a**) Dynamic monitoring of plasma cfDNA status; (**b**) Dynamic monitoring of urine cfDNA status. Different patients ID are defined in the legenda. plasma/urine PD, progressive disease defined as any increased in allelic frequency of the sensitizing EGFR mutation in relation with baseline levels on plasma or urine cfDNA.

**Table 1 cancers-13-05403-t001:** Patients’ baseline characteristics.

Patients’ Characteristics		No. (%)
Age, median (range)		52.5 (42–67)
Gender	male	7 (35%)
	female	13 (65%)
ECOG PS	0–1	19 (95%)
	2	1 (5%)
Smoking history	Nonsmoker	11 (55%)
	Former smokers/smokers	9 (45%)
Stage	IV	20 (100%)
Histology at diagnosis	Adenocarcinoma	20 (100%)
EGFR activating mutation	Ex19del	12 (60%)
	L858R	7 (35%)
	G719X	1 (5%)
First line EGFR TKI	Gefitinib	13 (65%)
	Erlotinib	4 (20%)
	Afatinib	3 (15%)
Second line EGFR TKI	Osimertinib	20 (100%)

**Table 2 cancers-13-05403-t002:** Resistance mechanisms to osimertinib and comparison between tissue biopsy and plasma specimens.

Patient	Plasma at PD	Tissue at PD
PLU-02	TP53	n.a.
PLU-03	TP53, NRAS, PIK3CA, KDR, EGFR amp	n.a.
PLU-04	EGFR T790M, EGFR C797S, MET amp, EGFR amp	n.a.
PLU-06	ROS1, SMAD4, FGFR2	n.a.
PLU-07	TP53, CDK4, EGFR amp, MET amp	MET amp
PLU-09	TP53, EGFR amp, ERRB2 amp	n.a.
PLU-10	SMAD4, KIT, EGFR amp	HER2 and EGFR polysomy
PLU-12	TP53, ROS1, EGFR amp, ERRB2 amp, MET amp	MET amp
PLU-13	EGFR T790M, EGFR C797S, TP53, ALK, EGFR amp	SCLC transformation
PLU-15	PIK3CA	EGFR ins 20
PLU-16	-	n.a.
PLU-17	EGFR amp, MET amp	n.a.
PLU-19	TP53, SMAD4, EGFR amp, MET amp	n.a.
PLU-21	-	MET amp

The table describes the putative resistance mechanisms that have been identified in plasma samples collected at PD to osimertinib. Only PD plasma samples that were defined as shedder on ddPCR test were considered. Shedder, patients with plasma positivity for EGFR activating mutation at the time of PD. Abbreviations: amp, amplification; n.a., not applicable; -, no molecular aberrations detected; PD, progressive disease.

**Table 3 cancers-13-05403-t003:** Review of the literature. The table shows some of the published studies on the detection of EGFR mutational status, conducted on non-blood liquid biopsy.

Study	Source	Cohort	Concordance with Tissue	Sensitivity and Specificity	Methods
S. Chen et al.	urine	150 paired tissue and urine samples	88% (90% L858R and L861Q 71%)	-	ddPCR
H. Yu et al.	urine	130 paired tissue and urine samples	83.1% (del19 and L858R)	-	ddPCR
K. L. Reckamp et al.	urine	63 paired tissue and urine samples	-	72% T790M (sensitivity) 96% T790M (specificity) 75% L858R (sensitivity) 100% L858R (specificity) 67% del19 (sensitivity) 94% del19 (specificity)	NGS
S. Ding et al.	saliva	68 paired plasma and saliva samples	83.78% (del19, L858R and T790M)	-	ddPCR
K. Hackner et al.	saliva	28 paired plasma and saliva samples	86% T790M 78% L858R 45% del19	-	ddPCR
F. Su et al.	saliva	37 paired tissue and saliva samples	-	90.9% EGFR mutational status (sensitivity) 96% EGFR mutational status (specificity)	ARMS-PCR
Nishii Kazuya et al.	EBC	21 EBC samples	-	27.3% L858R (sensitivity) 80% L858R (specificity) 30% del19 (sensitivity) 90.9% del19 (specificity) 22.2% T790M (sensitivity) 100% T790M (specificity)	ddPCR
Smyth’s et al.	EBC	10 paired EBC and plasma	-	not calculated	ddPCR

## Data Availability

The datasets that support the findings of this study are not publicly available in order to protect patients’ privacy. However, the data are available from the corresponding author on reasonable request.

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
