# Peer review of "Monitoring cfDNA in Plasma and in Other Liquid Biopsies of Advanced EGFR Mutated NSCLC Patients: A Pilot Study and a Review of the Literature"

_cancers, 2021, doi:10.3390/cancers13215403_

Round 1

Reviewer 1 Report

I really enjoyed reading this manuscript from Varze' and colleagues. I congratulate the authors for the clear presentation of the data and the extensive review of the literature. I recommend this work for publication.

Reviewer 2 Report

In my opinion authors did a good job and some novel findings were provided. I have only few minor comments, that should be clarified prior to publication.

  1. Please describe whether authors noticed any differences concerning patients' status such as gender and especially the previous treatment regimen.
  2.  According to the authors, how does the findings can improve the management of EGFR-mutated lung cancer?
  3.  Some cons of the work should be considered.
  4. The figures are quite unreadable, they should be modified or provided in better resolution. 

Reviewer 3 Report

This is a timely paper in the field of ctdna and lung cancer, in particular with the recent approval of KRAS targeted therapies, the detection of KRAS mutations routinely via tissue/blood is important. Moreover, the dynamic use of ctdna is promising for the field, in light of the egfr mutational monitoring studies. 

Specific comments:

  1. I strongly disagree with page 2, line 61 where the authors say that liquid biopsy accounts for both 'spatial and temporal tumour heterogeneity' - this has not been shown. How can liquid biopsy show spatial heterogeneity? Please see recent publications by David Rimm's lab and Monkman et al., cancers 2020 clearly showing that only tissue is able to do this. i would rephrase this.
  2. sample processing- how long was plasma processed post collection? It is important to clearly define these as standardisation protocols are being developed in the field and it shows that plasma needs to be processed under 4 hours post bleed.
  3. how was ctdna quantified?
  4. recent ctdna papers in lung cancer should be cited - especially those that track egfr/kras/pik3ca mutations - see Kulainghe et al., Lung cancer 2021
  5. table 2 - please define PD
